# Rescue of Low-Yield DNA Samples for Next-Generation Sequencing Using Vacuum Centrifugal Concentration in a Clinical Workflow

**DOI:** 10.3390/reports6020023

**Published:** 2023-05-23

**Authors:** Lau K. Vestergaard, Nicolai S. Mikkelsen, Douglas V. N. P. Oliveira, Tim S. Poulsen, Estrid V. Hoegdall

**Affiliations:** Molecular Unit, Department of Pathology, Herlev Hospital, University of Copenhagen, DK-2730 Herlev, Denmark

**Keywords:** vacuum centrifugation, low-yield DNA, linear regression models, next-generation sequencing, cancer, targeted therapeutics, clinical diagnosis

## Abstract

The implementation of next-generation sequencing (NGS) in clinical oncology has enabled the analysis of multiple cancer-associated genes for diagnostics and treatment purposes. The detection of pathogenic and likely pathogenic mutations is crucial to manage the disease. Obtaining the mutational profile may be challenging in samples with low yields of DNA—reflected by the type of biological material, such as formalin-fixed paraffin-embedded tissue (FFPE), needle biopsies, and circulating free/tumor DNA, as well as a sparse tumor content. Moreover, standardized strict procedures for the extraction of DNA in a clinical setting might contribute to lower amounts of DNA per µL. The detection of variants in low-yield DNA samples remains a challenge in clinical diagnostics, where molecular analyses such as NGS are needed. Here, we performed vacuum centrifugation on DNA extracted from five FFPE tissue blocks, with concentrations below 0.2 ng/µL. Through NGS analysis, we found that low-yield DNA samples could be concentrated to sufficient levels, without compromising the mutational profile.

## 1. Introduction

Genomic profiling using next-generation sequencing (NGS) is well implemented in clinical diagnostics for the analyses of multiple cancer-associated genes for diagnostic and treatment purposes. For instance, the proto-oncogenes *KRAS*, *BRAF,* and *NRAS* require a gain-of-function mutation to become tumorigenic [1]. *KRAS* mutations are frequently observed in colorectal cancer, whereas mutations in *BRAF* and *NRAS* occur at a lower rate [2,3]. The identification of these mutations is crucial for guiding therapy cancers targeting the RAS-RAF signaling pathway [4]. The detection of pathogenic and likely pathogenic mutations in tumor suppressor genes, such as *BRCA1* and *BRCA2,* in patients diagnosed with prostate cancer, ovarian cancer, or breast cancer is also employed in clinical interventions [5,6].

In situations where material or tumor content is sparse, these mutational identifications to guide therapy are challenging and may cause inconclusive results. Laboratories are standardized to fulfill strict procedures for the extraction of DNA in a clinical setting, such as specific volumes for sample dilution. Hence, that can lead to low amounts of DNA per µL. For molecular techniques, such as NGS, the DNA concentration needs to meet a defined threshold for reliable and optimal sequencing results. The yield of DNA is dependent on the size of the gene panel investigated, i.e., the larger a gene panel is, the more DNA is required as input material. DNA levels below the limits may translate directly into impacting the performance of the NGS. In situations where DNA concentrations are below the recommendations defined by the manufacturer, the application of vacuum centrifugation may be beneficial to obtain an increased concentration. A study by Sánchez and coworkers showed that applying vacuum centrifugation can concentrate DNA samples and may not affect DNA integrity [7].

The detection of malignancies in the early stages of disease increases overall survival [8]. In that perspective, some healthcare programs, e.g., for breast cancer or prostate cancer, include needle biopsies concerning screening programs. This means that sparse tumor tissue may be available for DNA extraction.

Formalin-fixed and paraffin-embedded (FFPE) tissue is widely used in clinical routine settings due to its broad application in molecular techniques for the molecular characterization of tumors. However, it may be a challenge to obtain enough DNA for downstream analysis. The chemical properties of formalin fixation have an impact on DNA integrity, and an increase in the amount of DNA degradation is commonly observed in FFPE tissue [9]. DNA quality and quantity are correlated to the time a sample is exposed to formalin, i.e., the longer the sample is embedded in FFPE, the more DNA degradation and cytosine deamination may be observed [9]. NGS may fail more often when DNA extracted from older FFPE blocks is used in the downstream analysis [10,11].

The use of FFPE causes the risk of cytosine deamination, which causes a transition of C:G > T:A, thus introducing false positives into the downstream interpretation of variants during analysis. The treatment of DNA extracted from FFPE tissue with the uracil DNA glycosylase has been shown to significantly reduce the amounts of false positives occurring from cytosine deamination [12,13,14]

It remains a challenge to deal with clinical diagnostics in low-yield DNA patient samples where molecular analyses such as NGS are needed. Information about the underlying molecular profile is crucial to guide targeted treatment options. To our knowledge, no studies describing vacuum centrifugation of low-yield DNA samples followed by downstream analysis with NGS have been reported. Such information to deal with incompatible DNA samples would be helpful in a clinical routine setting and of importance to the possibility of still guiding treatment decisions in cases with limited amounts of DNA. Herein, we investigate the potential of applying vacuum centrifugation on a total of 11 DNA samples from five cancer patients with DNA levels insufficient for NGS analyses.

## 2. Materials and Methods

### 2.1. Samples

A total of five FFPE tissue specimens were retrospectively collected from three patients diagnosed with colorectal cancer, one patient diagnosed with prostate cancer, and one diagnosed with head and neck cancer.

### 2.2. Vacuum Centrifugation

Samples were vacuum-centrifugated using the SpeedVac™ DNA130 Vacuum Concentrator (Thermo Fisher Scientific, Waltham, MA, USA). All vacuum centrifugations were performed at room temperature (22–24 °C), at various time points.

### 2.3. Data Handling and Statistical Analysis

To find the linear relationship, we started with one of the samples and diluted it into two batch solutions, with a criterion of the concentration being below 1 ng/μL. This would mimic the samples that are excluded in our clinical setting for NGS analysis. The sample was diluted into concentrations of 0.746 ng/μL and 0.170 ng/μL. Four replicates with a volume of 55 μL of the batch solutions were made for the following time points: 5, 10, 15, 20, 25, and 30 min for the 0.746 ng/μL batch and 10, 20, 30, and 40 min for the 0.170 ng/μL batch. Both the concentration and volume of samples were measured post-vacuum centrifugation (Appendix A).

Standard linear regression (1) was performed to describe the relationship between the dependent variables (concentration and volume) as a function of the independent variable (time). The standard linear regression models for concentration and volume are described in Equations (2) and (3).
Y_i_ = β_0_ + β_1_ X_i_(1)
Y_concentration_ = β_intercept_ + 0.02624 X_concentration_(2)
Y_volume_ = β_intercept_ − 1.09675 X_volume_(3)

The variance (4) and standard deviation (SD) (5) were calculated as described below:(4)$$\sigma^{2}=\overset{n}{\underset{i=1}{\sum }}\frac{{\left(x_{i}-\bar{x}\right)}^{2}}{N}$$
(5)$$\sigma =\sqrt{\overset{n}{\underset{i=1}{\sum }}\frac{{\left(x_{i}-\bar{x}\right)}^{2}}{N}}$$

To verify the relationships, four new dilutions were made spanning from the lowest dilution of 0.294 ng/μL to the highest being 1.212 ng/μL. All samples were diluted in 55 μL and vacuum-centrifugated for 20 min. Concentrations and volumes were measured post-vacuum centrifugation (Appendix A). We simulated 2000 new data points for each dilution using a Gaussian distribution with three standard deviations from the initial data points measured to model sample concentration (ng/μL) and sample volume (μL). The three SD are representative of eventual measurement uncertainties.

All data handling, data simulation, and statistical analysis were performed using Python programming language (v.3.7) and SciPy (v. 1.6.3). Graphics were created using the Python packages seaborn (v.0.11.2) and Comut (v.0.0.3) [15].

### 2.4. DNA Extraction

Areas of cancerous tissue were identified and marked by specialized pathologists. Slides used as reference points for the extraction of cancerous areas were stained with hematoxylin and eosin (H&E). Tissue for DNA extraction was isolated using a 1 mm disposable puncher. Extraction and purification of genomic DNA were conducted using Maxwell^®^ RSC DNA FFPE Kit (Promega, Madison, WI, USA) following the manufacturer’s instructions. The DNA concentration was determined using the Qubit™ ds DNA High-Sensitive Assay Kit (Thermo Fisher Scientific, Waltham, MA, USA) on the Qubit fluorometer (Thermo Fisher Scientific, Waltham, MA, USA) according to protocol. Samples with a concentration below 0.5 ng/mL were measured with an input volume of 4 µL DNA and 196 µL Qubit working solution. Patient 4 and patient 5 had DNA extracted twice as initial DNA extractions yielded insufficient DNA concentrations to complete NGS analysis in a routine setting. DNA was diluted in a volume of 55 μL due to manufactory instruction. Concentrations of pre- and post-vacuum centrifugation can be found in Appendix A.

DNA was extracted within an average of 52 days from the sample that underwent the FFPE protocol. DNA integrity was not assessed before NGS analysis due to the relatively limited period of storage in FFPE.

### 2.5. DNA Library Preparation and Sequencing

We employed the Oncomine™ Focus Assay (OFA) (Thermo Fisher Scientific, Waltham, MA, USA), a panel comprising 52 cancer-associated genes, and the Oncomine™ Comprehensive Assay v3 (OCA) (Thermo Fisher Scientific, Waltham, MA, USA), comprising 161 genes. Before library preparation, DNA was treated with Uracil-DNA Glycosylase (Thermo Fisher Scientific, Waltham, MA, USA) to minimize the effects of cytosine deamination caused by FFPE preparation.

Manufacturer recommends using 1–10 ng gDNA for OFA (MAN0015819, C.0) and 10 ng gDNA for OCA (MAN0015885, C.0) as input for multiplex amplicon-based PCR amplification. Briefly, a PCR amplicon-based enrichment strategy includes several benefits, such as a low requirement of input DNA, application to FFPE, and the ability to reach greater sequencing depth [16,17]. A qPCR protocol was used as a quality control step to assess the performance of the multiplex amplicon-based PCR and, indirectly, the DNA. Moreover, these measurements were used to adjust the concentration of the libraries to 50 pM for sequencing.

Library preparation was finalized using the Ion Chef™ System (Thermo Fisher Scientific, Waltham, MA, USA), and samples were loaded onto Ion 550™ chips (Thermo Fisher Scientific, Waltham, MA, USA).

The sequencing was performed using the Ion Torrent sequencing technology on the Ion S5™XL Sequencer (Thermo Fisher Scientific, Waltham, MA, USA) following the manufacturer’s instructions. A clear description of the technology is outlined in detail in a previous paper published by our group [18]. Sequencing data were initially pre-processed, aligned, and analyzed in Ion Reporter™ Software (v.5.18.4.0).

The human genome assembly 19 (hg19) was used as a reference for alignment. Preliminary filtering of variants was conducted using the Oncomine Variants (5.18) filter embedded in the Ion Reporter™ Software. Variants were subsequently subjected to a second round of filtering using an in-house-developed algorithm. The properties of this filtering are outlined and described in a recent study performed in our research setting [19]. In our routine clinical setting, the cut-off value for allele frequency is 5%. Moreover, rescue filtering with a cut-off of 3% is applied to capture putatively true variants. These variants are extensively quality checked and validated to confirm whether they are true or artificially introduced.

All variants were checked and annotated using the ClinVar database (v.20221030) [20].

OCA allows for the detection of single-nucleotide variants (SNVs), multiple-nucleotide variants (MNVs), and small insertions/deletions (indel). The OCA panel has, since 2017, been implemented to assist oncologists in therapeutic decision making in our clinical setting. The performance was recently assessed and used to focus on treatment options for refractory metastatic colorectal cancer [21]. Hence, variant detection was not investigated with an orthogonal method such as ddPCR.

## 3. Results

### 3.1. Prediction of Concentration and Volume

To predict the interval a sample would lie within after vacuum centrifugation, we applied standard linear regression models for both DNA concentration and volume. The slope value for the concentration was 0.02624, whereas the slope value for the volume was −1.09675.

We simulated 2000 new data points for four samples, with a Gaussian distribution and three SD. The SD for the concentration was 0.03687 ng/μL and the SD for the volume was 0.94166 μL. After 20 min of vacuum centrifugation, the samples were displayed within three SDs (Figure 1A). The sample with a starting concentration of 0.294 ng/μL appeared to be at the border of the three SDs. Moreover, 2000 new data points for the volume with a Gaussian distribution and three SD were simulated. After 20 min of vacuum centrifugation, the samples had a volume of 32 μL or 32.50 μL and a predicted volume of 33.06 μL (Figure 1B).

A total of 11 DNA samples from five FFPE tissues were subjected to NGS after vacuum centrifugation. The concentrations of the 11 samples were plotted post-vacuum centrifugation along with the relationship for concentration as a function of time (minutes) and are shown in Figure 2. All concentrations were above 1 ng/μL.

### 3.2. Molecular Profiling

Targeted sequencing with OFA and OCA was performed on 11 DNA samples from five cancer patients. All 11 samples were subjected to NGS. The original sample from where dilutions were made had previously been sequenced, and the mutational profile was known—this included samples S1.0, S2.0, S3.0, S4.0, and S5.0 (Figure 3). These samples were initially selected based on sufficient DNA concentration for NGS analysis (10 ng). In total, 34 true variants were identified, with all being missense mutations. By intra-comparison, we found that the same variants were identified, with coverage and allele frequencies within the same ranges, independently of the initial DNA concentration. A complete list of the variants identified, including their coverage and allele frequencies, is outlined in Appendix A.

The mutational profile of pathogenic variants and variants of uncertain significance is shown in Figure 3. The allele frequencies for the identified variants are depicted in Figure 3. Allele frequencies for samples of patient 1, patient 3, and patient 4 were almost identical, with a small, calculated variance (σ_patient1_*KRAS*_^2^ = 0.664, σ_patient1_*EGFR*_^2^ = 1.901, σ_patient3_*BRAF*_^2^ = 0.280, σ_patient3_*PIK3CA*_^2^ = 0.016, σ_patient4_*TP53*_^2^ = 4.796). Allele frequencies found in samples from patient 5 showed larger variances in identified allele frequencies (σ_patient5_*TP53*_^2^ = 60.840, σ_patient5_*CREBBP*_^2^ = 20.839, σ_patient5_*BRCA2*_^2^ = 27.405, σ_patient5_*BRCA2*_^2^ = 27.772, σ_patient5_*BRCA2*_^2^ = 18.147, σ_patient5_*NOTCH1*_^2^ = 75.690, σ_patient5_*PTCH1*_^2^ = 60.372, σ_patient5_*ERBB4*_^2^ = 12.145, σ_patient5_*MYCN*_^2^ = 13.286, σ_patient5_*MYCN*_^2^ = 13.395). For this patient, the H&E-stained tissue slide of the tumor was checked to estimate tumor percentage. The slide that raised the lower allele frequencies showed a lower tumor percentage compared to the second slide used for DNA extraction.

To confirm whether random mutations could be introduced into a sample and create noise for analysis, we included samples of negative control (Patient 2). Initially, sample S2.0 did not harbor true variants within the 52 genes comprising the OFA panel. The results showed that vacuum centrifugation did not introduce artificial true variants in samples of the negative control or the other samples.

## 4. Discussion

The demand for NGS analyses using different sources of biological material increases rapidly for clinical management to assist oncologists in guiding therapy. However, DNA concentrations may be too low to meet the requirements to conduct proper molecular analyses. In the present study, we used samples with DNA concentrations as low as 0.176 ng/μL for investigating the possibility of using vacuum centrifugation to concentrate samples with an initial low yield of DNA. Furthermore, we employed NGS to determine whether the molecular profile remains intact upon vacuum centrifugation. We found that using a simple linear regression model, with three standard deviations, we can describe the relationship between the dependent values, concentration and volume, and the independent value, time.

When comparing the mutational profiles obtained from NGS analysis using either OFA or OCA, we found that using up-concentrated DNA samples provided similar results compared to the original results from samples not subjected to vacuum centrifugation. When comparing the intra-allele frequencies obtained from the patients’ NGS analysis, a good correlation is observed, with a small variance in allele frequency in patients 1, 2, 3, and 4. However, we observed that samples from patient 5 harbored different allele frequencies.

This variance in allele frequencies may originate from DNA being extracted from two different layers of the tumor, causing the tumor cell composition to be different from each other. Via inspection of the H&E-stained tissue slide of the tumor, the first DNA extraction harbored less tumor compared to the material used in the second DNA extraction. The tumor cell content may explain the difference in the allele frequency. Despite this, our results indicate that a DNA concentration initially below the threshold for NGS analysis may be suitable for NGS after vacuum centrifugation. We investigated samples from three different cancer types with the same results. This suggests that vacuum centrifugation might not only be limited to tissue only but may also be used for circulating tumor DNA/circulating free DNA from plasma samples and other liquid biopsies, such as ascites and peritoneal fluids, where DNA concentrations are known to be challenging for molecular analyses [22].

Other library preparation protocols to handle samples with low concentrations of DNA for targeted panel sequencing or whole-exome sequencing are available. The verification and comparison of these methods to vacuum centrifugation will contribute to extended knowledge for handling samples with low-yield DNA samples and these make up a relevant study to conduct in the future. Vacuum centrifugation is an easily applicable method that is independent and compatible with all library preparation protocols and sequencing technologies.

## 5. Conclusions

In conclusion, samples with a low yield of DNA can be up-concentrated to sufficient levels for NGS analysis. Here, we showed that samples with a concentration below 0.2 ng/µL can be concentrated to sufficient DNA yield through vacuum centrifugation. NGS analysis revealed complete overlap in the identified variants, thereby resulting in conclusive mutational results for clinical treatment management. Vacuum centrifugation has already been introduced into our clinical routine setting to handle challenging samples with initial low concentrations of DNA.

## Figures and Tables

**Figure 1 reports-06-00023-f001:**
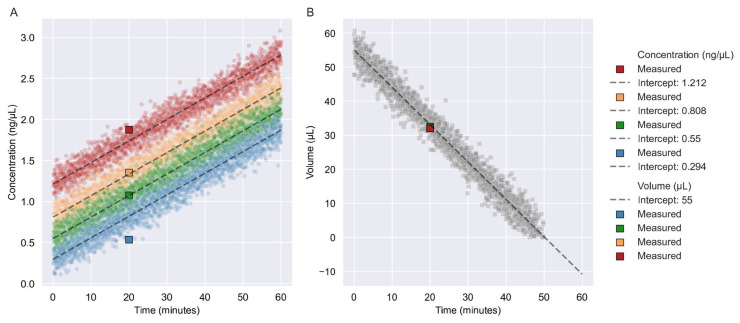
Verification of linear regression models. (**A**) Here, 2000 new data points were simulated using the linear relationship model with a slope value of 0.02624 and a Gaussian distribution with three SDs. Each simulation is depicted with a color for each of the four concentrations: 1.212 ng/μL, 0.808 ng/μL, 0.550 ng/μL, and 0.294 ng/μL. Intercepting values are represented by the initial concentration of the sample. (**B**) 2000 new data points were simulated using a linear relationship model with a slope value of −1.09675 with a Gaussian distribution and three SD. The intercept value was the starting volume of the dilution of 55 μL. NOTE: Simulated data points are represented as circles, whereas squares are representative of the verification samples. Dashed lines indicate the slope of the relationships.

**Figure 2 reports-06-00023-f002:**
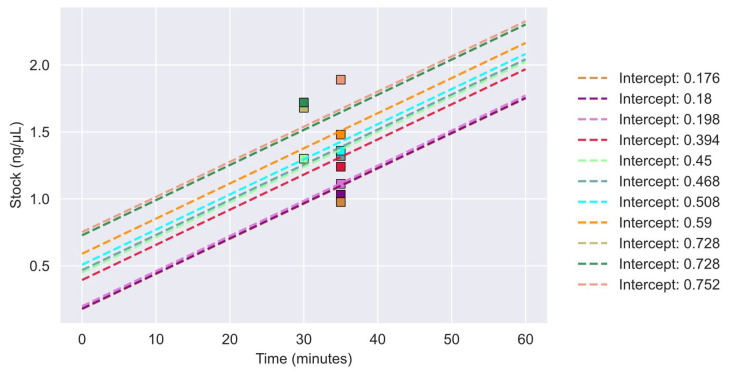
Concentrations of vacuum-centrifugated samples for NGS analysis. Sample concentrations are plotted with their respective linear regression model. Each final sample concentration after vacuum centrifugation is depicted as a colored square. The sample linear regression models are represented as dashes lines in corresponding colors to the squares. Intercept values are represented by the initial concentration of the sample.

**Figure 3 reports-06-00023-f003:**
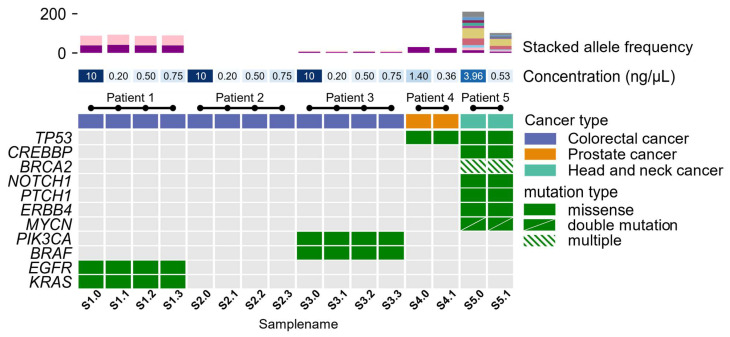
A co-mutational plot of 11 mutated genes in five patients. The top panel shows the allele frequency of identified variants in the samples. Mid-panel designated concentration (ng/μL) shows the initial start concentration of the sample ranging between 0.176 ng/μL and 0.752 ng/μL and 10 ng/μL. Samples with a concentration below 1 ng/μL were subjected to vacuum centrifugation. The mid-section colored row is indicative of the patient’s cancer type, involving colorectal cancer (purple), prostate cancer (orange), and head and neck cancer (turquoise). All identified variants of either pathogenic or variants of unknown significance are depicted in the bottom panel. Green-colored squares are representative of mutated genes, while gray squares indicate non-mutated genes within the sample. Diagonal, divided, green-colored squares denote those genes with two mutations within a gene. Shaded green-colored squares denote those genes with three or more mutations within a gene.

## Data Availability

Due to sensitive information and the EU data protection legislation, we are unable to disclose the next-generation sequencing datasets for this study, unfortunately. Nonetheless, if a researcher has an interest in our data, they are welcome to contact us and collaborate. The data that support the findings of this study can be requested from The National Secretariat for Bio- and Genome Bank Denmark, RBGB.sekretariat.herlev-og-gentofte-hospital@regionh.dk, Herlev Hospital, Borgmester Ib Juuls Vej 73, 2730 Herlev, Denmark.

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
