# Peer review of "Rescue of Low-Yield DNA Samples for Next-Generation Sequencing Using Vacuum Centrifugal Concentration in a Clinical Workflow"

_reports, 2023, doi:10.3390/reports6020023_

Round 1

Reviewer 1 Report (Previous Reviewer 2)

Dear authors, 

the manuscript is really changed and follows all the suggestions done on the previously revision.
I believe that in this way the manuscript can be suitable for publication on Reports. 

Author Response

Reviewer 2 Report (New Reviewer)

In the communication entitled "Rescue of Low Yield DNA Samples for Next-generation Sequencing using Vacuum Centrifugal Concentration in a Clinical Workflow" by Lau K. Vestergaard et al., the authors aimed to address the challenge of detecting variants in low yield DNA samples intended for NGS testing, by performing vacuum centrifugation on DNA extracted from five FFPE tissue blocks. The authors found that samples with a low yield of DNA could be concentrated to reach sufficient levels for NGS analysis. That analysis showed agreement in the detection of variants between concentrated samples and their original counterpart.

The study, albeit using a very small number of samples, is very straightforward. However, it would be desirable that the authors address some revisions in order to consider publication:

1-      The paragraph in red font on page 2, needs to be revised for formatting issues (lack of spaces between words).

2-      The manuscript is somewhat well written, however, grammar revisions by a native English-speaking person are strongly recommended.

3-      It would be desirable that the authors provide the variant allele frequency (VAF) of the detected variants in the relevant genes obtained prior and after centrifugation for both assays, OFA and OCA. Did the authors choose samples with variants at, or near, the limit of detection (LoD) of these assays? Such a choice would had allowed for a better assessment of the impact of DNA concentration on the LoD of these assays.

4-      On the same line of thought, it would be desirable that the authors provide the coverage (mean depth) obtained for each sample before and after centrifugation for both assays, OFA and OCA.

The manuscript is somewhat well written, however, grammar revisions by a native English-speaking person are strongly recommended.

Author Response

Reviewer 3 Report (New Reviewer)

In this article, the authors attempted to increase the yield of specimens' DNA through vacuum centrifugation.

Readers and I would like to know how centrifugation can decrease the purity of DNA. A bioanalyzer of other images of DNA should be presented.

What was the NGS sequencing library preparation type? Does it include PCR steps? This needs to be described because the inhibitors in concentrated DNA can significantly affect the quality of library products.

Round 2

Reviewer 2 Report (New Reviewer)

None.

Reviewer 3 Report (New Reviewer)

no suggestions, this is second review

This manuscript is a resubmission of an earlier submission. The following is a list of the peer review reports and author responses from that submission.

Round 1

Reviewer 1 Report

 Vestergaard L et al described very well the need to use high quality DNA to detect mutational status by NGS. Although, NGS techniques require less DNA amount than the other high-troughput methods, some samples don't achieve the minimum yield required, as FFPE tumor biopsies. Vacuum centrifugation seemed to be a good option to increase DNA amount. In this paper, the author showed the advantages of the vacuum centrifugation both in  sustained dilutions and in NGS analysis of the same samples. 

I think the the major pitfall of the study is the limited number of sample group. The author should increase the number of tested samples (almost 20). Moreover, no data were available about the quality of starting material (starting concentration, integrity of DNA, etc).  these information should improve the quality of the paper.

Reviewer 2 Report

The manuscript " Rescue of Low Yield DNA Samples for Next-generation Sequencing using Vacuum Centrifugal Concentration in a Clinical Workflow " by Vestergaard LK et al., report is a well-written paper on the use of vacuum centrifugal concentration on low yield DNA samples additional experiments needed

Major revisions are required, listed below.

The authors apply vacuum centrifugal concentration to DNA samples extracted from FFPE tissue and the results obtained are very interesting and useful for using samples with low DNA concentration for NGS.

The authors use a correct approach and it is well described.

The processed samples are FFPE samples which as the main problem, in addition to that of the initial concentration, are very degraded and this is the main problem that prevents obtaining good quality NGS results. The authors did not consider this aspect and did not check the degradation status of the sample.

How long have samples been included in FFPE? The longer they are in FFPE the higher the state of degradation.

There are library prep protocols that start from very low amounts of DNA applied for panel or WES (ie fetal cell-free DNA or ancient DNA) sequencing, the authors should apply these protocols (which do not involve vacuum centrifugal concentration) and verify if there are significant differences.

The authors indicate that sequence variants were identified in the samples subsequently sequenced, were the variants confirmed with an alternative method? I suggest the authors confirm by dd-PCR for the validation of the results being both samples with a low concentration of DNA and samples in FFPE.